# The Roles of CCCH Zinc-Finger Proteins in Plant Abiotic Stress Tolerance

**DOI:** 10.3390/ijms22158327

**Published:** 2021-08-03

**Authors:** Guoliang Han, Ziqi Qiao, Yuxia Li, Chengfeng Wang, Baoshan Wang

**Affiliations:** Shandong Provincial Key Laboratory of Plant Stress, College of Life Science, Shandong Normal University, Ji’nan 250014, China; adg129@126.com (Z.Q.); Liyx202103@163.com (Y.L.); wywcf123@163.com (C.W.)

**Keywords:** abiotic stresses, CCCH zinc-finger proteins, plants, regulation pathways, transcription factor

## Abstract

Zinc-finger proteins, a superfamily of proteins with a typical structural domain that coordinates a zinc ion and binds nucleic acids, participate in the regulation of growth, development, and stress adaptation in plants. Most zinc fingers are C2H2-type or CCCC-type, named after the configuration of cysteine (C) and histidine (H); the less-common CCCH zinc-finger proteins are important in the regulation of plant stress responses. In this review, we introduce the domain structures, classification, and subcellular localization of CCCH zinc-finger proteins in plants and discuss their functions in transcriptional and post-transcriptional regulation via interactions with DNA, RNA, and other proteins. We describe the functions of CCCH zinc-finger proteins in plant development and tolerance to abiotic stresses such as salt, drought, flooding, cold temperatures and oxidative stress. Finally, we summarize the signal transduction pathways and regulatory networks of CCCH zinc-finger proteins in their responses to abiotic stress. CCCH zinc-finger proteins regulate the adaptation of plants to abiotic stress in various ways, but the specific molecular mechanisms need to be further explored, along with other mechanisms such as cytoplasm-to-nucleus shuttling and post-transcriptional regulation. Unraveling the molecular mechanisms by which CCCH zinc-finger proteins improve stress tolerance will facilitate the breeding and genetic engineering of crops with improved traits.

## 1. Introduction

Throughout their lives, plants are exposed to various complex environmental conditions [1,2,3,4,5]. Environmental conditions that are not conducive to plant development and growth are collectively referred to as stress [6,7,8]. Generally speaking, stresses include biotic stresses (bacteria, viruses, and insect pests) and abiotic stresses (drought, high and low temperatures, high salt levels, waterlogging and heavy metals) [8,9,10,11]. Abiotic stresses can seriously decrease the growth and yield of crops and are important factors limiting crop yields worldwide [6,12,13,14]. Salt and drought stresses affect 10% of the world’s arable land, reducing global crop yields by more than 50%, whereas biotic stresses such as pests and diseases also result in substantial crop losses [15,16,17].

To adapt to variable environmental conditions, plants have evolved diverse regulatory pathways to receive and respond to different stress signals [18,19]. When a plant is subjected to stress, it activates the corresponding regulatory pathway; the resulting responses help it to survive under the stress [7,20,21,22]. In harsh conditions such as high salt levels, extreme drought and high concentrations of heavy metals or arsenic, halophytes [23,24], xerophytes [25,26] and arsenic hyperaccumulators [27,28] have formed their own unique morphology that resists abiotic stress during long-term evolution. Plant responses to abiotic stresses involve a large number of transcription factors [29,30,31,32]. Results from genome-wide sequencing, differential transcriptome analysis, and functional analysis of numerous genes have shown that the transcription factor families involved in abiotic stress tolerance in plants include bHLH (basic helix-loop-helix) [33,34,35], WRKY [10,36,37,38], bZIP (basic leucine zipper) [39,40,41], homeodomain [42,43], HSF [3,44,45,46], NAC [47,48,49], MYB [50,51,52], MADS-box [53,54,55], AP2/ERF [56,57,58], and zinc-finger proteins [59,60,61]. The large and diverse zinc-finger protein family plays important roles in all aspects of plant growth and development. The first zinc-finger protein (TF IIIA) involved in transcriptional regulation was found in *Xenopus laevis* oocytes in 1985; since then, zinc fingers with a variety of functions have been found in animals, plants, yeasts, and viruses [62]. 

Plant genomes encode large numbers of CCCH zinc-finger proteins. For example, genomic surveys identified 68 CCCH zinc-finger protein genes in *Arabidopsis thaliana* (L.) Heynh [63], 67 in *Oryza sativa* L. [63], 68 in *Zea mays* L. [64], 91 in *Populus trichocarpa* Torr. & Gray [65], 34 in *Medicago truncatula* Gaertn. [66], 36 in *Aegilops tauschii* Coss. [67], 80 in *Solanum lycopersicum* L. [68], 69 in *Vitis vinifera* L. [69], 58 in *Cicer arietinum* L. [70], and 103 in *Brassica rapa* L. ssp. Pekinensis [71]. CCCH zinc-finger protein genes participating in many biological functions such as development and growth have been cloned and studied in leaf senescence in rice (*OsDOS*) [72], seed germination in Arabidopsis (*AtTZF4*, *AtTZF5*, and *AtTZF6*) [73], flowering time in *Medicago sativa* L. (*MsZFN*) [74], cell elongation in Arabidopsis (*AtC3H14*) [75], secondary xylem formation in poplar (*PdC3H17* and *PdC3H18*, a homolog of *AtC3H14*) [76], seed storage in rice (*OsGZF1*) [77], anther development and secondary wall thickening in poplar (*C3H14* and *C3H15*) [78], natural rubber biosynthesis in *Hevea brasiliensis* (*Willd. ex A. Juss.*) *Müll. Arg.* (*HbCZF1*) [79], and *oleic acid homeostasis in Brassica napus* L. (*BnZFP1*) [80]. In addition to these processes, CCCH zinc-finger proteins play key roles in regulating the tolerance of plants to abiotic and biotic stresses [81]. Here, we focus on their role in plant responses to abiotic stress.

## 2. Domain Structure of CCCH Zinc-Finger Proteins

A characteristic feature of zinc-finger proteins is the zinc-finger domain, which forms a finger-shaped tetrahedral structure by binding zinc ions, and is an important domain in eukaryotic transcription factors [82,83]. Based on their structural features, zinc-finger domains can be divided into several different categories [84]. The most common zinc fingers in transcription factors are C2H2-type or CCCC-type, named after the configuration of cysteine (C) and histidine (H) in the zinc finger domain [85]. Compared with other types of zinc finger proteins, CCCH zinc fingers are less common, accounting for about 0.8% of all zinc fingers [86,87], and these have their own specific biological functions.

The CCCH zinc-finger proteins of plants have one to six copies of the conserved CCCH motif [88]. CCCH motifs vary in the number of amino acid residues separating the conserved Cys and His residues. The consensus sequence was originally defined as C-X_6–14_-C-X_4–5_-C-X_3–4_-H [89], but a detailed analysis of CCCH zinc-finger motifs in Arabidopsis and rice identified five CCCH motifs not mentioned in the original classification: C-X_4_-C-X_5_-C-X_3_-H, C-X_5_-C-X_4_-C-X_3_-H, C-X_7_-C-X_6_-C-X_3_-H, C-X_8_-C-X_6_-C-X_3_-H, and C-X_15_-C-X_5_-C-X_3_-H. Consequently, the motif has been redefined as C-X_4–15_-C-X_4–6_-C-X_3–4_-H [63]. The same analysis revealed that CCCH zinc-finger proteins containing the C-X_7–8_-C-X_5_-C-X_3_-H motif in the middle of the protein are the most prevalent, suggesting that this motif may be the original from which the others were derived [65]. With continuous research on CCCH zinc-finger proteins in different plants, it was found that the same pattern is present in other model plants [71]. Furthermore, the same study also found that the CCCH domain of CCCH zinc-finger proteins plays a key role in plant resistance [90]. 

A protein containing a tandem CCCH zinc-finger (TZF) motif is called a TZF protein [91]. Generally speaking, the TZF motif in animals is composed of two CCCH motifs (C-X_7_-_8_-C-X_5_-C-X_3_-H) separated by 18 amino acids [92]. However, it is variable in plants compared with animals. For example, TZF protein AtTZF1 in Arabidospsis contains two different motifs, C-X_7_-C-X_5_-C-X_3_-H and C-X_5_-C-X_4_-C-X_3_-H, separated by 16 amino acids [75,93], while in cotton, GhZFP1 also contains two different motifs, C-X_8_-C-X_5_-C-X_3_-H and C-X_5_-C-X_4_-C-X_3_-H zinc, separated by 16 amino acids [84]. Many TZF proteins have a plant unique arginine-rich (RR) region in front of the TZF motif, called RR-TZF, and the function of the RR-TZF family has been more deeply identified and studied in many model plants, including Arabidopsis and rice [94]. Additionally, zinc-finger proteins that do not have tandem motifs are called non-TZF proteins [95]. To date, TZF CCCH zinc-finger genes have been more extensively studied than non-TZF protein genes in plants [81,91].

In addition to the above structures, CCCH zinc-finger proteins have some other domains that are compatible with their biological functions. Many CCCH proteins contain a N- or C-terminal nuclear export signal sequence (NES) and/or nuclear localization sequence (NLS), which set their subcellular localization [84,96], as described below. Moreover, many CCCH zinc-finger proteins can activate transcription in Arabidopsis protoplasts or yeast systems, and domains involved in transcriptional activation have been identified at different positions within their protein sequences, such as the N terminus of the protein [90]. Furthermore, the N terminus of CCCH zinc-finger proteins is highly conserved among plants and can bind to DNA and RNA [97]. Some CCCH proteins participate in RNA metabolism and many of these carry RNA recognition motifs (RRMs) [98]. The different domain structures of CCCH proteins will be described in more detail in the below section, where we discuss protein localization and function.

## 3. Subcellular Localization of CCCH Zinc-Finger Proteins

To function, proteins must be localized to the appropriate part of the cell; in turn, a protein’s location in a cell can offer clues about its function [99,100]. So far, CCCH zinc-finger proteins show different patterns of localization: some are located in the nucleus, such as GhZFP1 [72], AtTZF11 [101], OsDOS [84], AtZFP1 [21], KHZ1 and KHZ2 [102], SAW1 [103] and OsC3H10 [104]. Some are located in the cytoplasm plasma membrane, such as Oxidation-related Zinc Finger 1 (AtOZF1) [105] and AtOZF2 [106]. Some are located in the cytoplasm, such as ZFP36L3 [107], ZC3H12a [108], and AtTZF2/3 [109], while some can shuttle between the cytoplasm and the nucleus, such as AtTZF1, AtTZF4, AtTZF5, AtTZF6, AtTZF7, and OsLIC [73,93,110,111]. A previous study found that the CCCH zinc-finger proteins ZAP, TTP, CMG1, and TIS11D in animals shuttle between the cytoplasm and the nucleus mainly through their N- or C-terminal NESs and NLSs. The leucine-rich NES also interacts with the nuclear export receptor CRM1 in the nuclear pore complex [96,112,113]. NESs are present in 54 CCCH proteins in Arabidopsis, including the whole 11 proteins of the IX subfamily, and the proteins such as AtTZF1, AtTZF2, AtTZF3, AtTZF4, AtTZF5, AtTZF6, AtTZF7 and AtTZF11 mentioned above were the representative members [63]. The widespread presence of shuttle signals in plant CCCH proteins suggests that they may be intra- and extranuclear shuttle proteins with important roles in signal transduction and stress responses [65]. In addition, AtTZF1, AtTZF4, AtTZF5, AtTZF6 and OsC3H10 co-localize with processing bodies (PBs) and stress granules (SGs) in the cytoplasm [104]. As aggregates of cytoplasmic messenger ribonucleoprotein complexes, PBs and SGs play important roles in the post-transcriptional regulation of genes and have been highly conserved during evolution. PBs and SGs play an important role in plant tolerance to abiotic stress [97]. AtTZF1, AtTZF4, AtTZF5, and AtTZF6 may function through a mechanism associated with PBs and SGs, and this function may be related to the cytoplasmic shuttle characteristics of these CCCH proteins [104,114].

## 4. Transcriptional and Post-Transcriptional Regulation by CCCH Zinc-Finger Proteins

### 4.1. Transcriptional Regulation

Many CCCH zinc-finger proteins are important transcriptional regulators, and their nuclear localization is consistent with this function [21,115]. Experiments have found that CCCH-type zinc-finger proteins such as AtTZF1 [93] and PEI1 (AtTZF6) [73,116] in Arabidopsis, and SAW1 [103] and Leaf and Tiller Angle Increased Controller (OsLIC) [110] in rice can bind to DNA in vitro. Conserved domains in CCCH zinc-finger proteins can bind to DNA and enable the zinc-finger protein to regulate downstream genes [77,110].

CCCH zinc-finger proteins can function as activators of transcription [102]. For example, the OsLIC protein of *Oryza sativa* has transcriptional activation activity in a yeast system, and a typical conserved EELR domain in the C terminus of the protein plays a key role in transcriptional activation [110]. Rice Early heading date 4 (Ehd4) also has detectable transcriptional activation activity in yeast. Deletion of the C-terminal region significantly decreases this activity, and it is speculated that the transcriptional activation domain of Ehd4 is near the C terminus [115]. By contrast, the transcriptional activation domains of some CCCH zinc-finger proteins seem to be N-terminal. For example, the region of AtC3H17 that shows the strongest transcription activation activity is a segment from 1 to 288 amino acids at the N terminus, and conserved glutamate residues in an EELR-like motif at the N terminus of AtC3H17 play a key role in transcriptional activation [117]. Under abiotic stress, these CCCH zinc-finger proteins regulate plant tolerance by directly activating the expression of related target genes. For example, AtZFP1 improves the salt tolerance of plants by activating downstream genes related to salt tolerance, such as *SOS1*, *AtGSTU5*, and *AtP5CS1* [21]. AtC3H17 improves the salt and oxidative tolerance of plants by regulating ABA-dependent responsive genes *RAB18*, *COR15A*, and *RD22* [95].

In addition to transcriptional activators, some other proteins are transcriptional repressors. In plants, GLUB-1-BINDING ZINC FINGER 1 (OsGZF1) from rice regulates the accumulation of gluten via transcriptional inhibition of *GluB-1* by binding to GCN4 (TGAGTCA) and PROL (TGCAAAG) motifs in the promoter region of the *GluB-1* gene [77]. Additionally, ILA1-interacting protein 4 (IIP4) from rice negatively regulates the expression of MYB61 and CESAs in secondary wall synthesis [118].

### 4.2. Post-Transcriptional Regulation

In addition to transcriptional regulation, CCCH zinc-finger proteins also participate in post-transcriptional regulation by binding to mRNA [71]. Many plant CCCH zinc-finger proteins have the ability to bind to RNA in vitro, such as AtTZF1 [93], KHZ1 and KHZ2 [102] in Arabidopsis. The human CCCH zinc-finger proteins TTP and BRF1 recognize the AU-rich elements (AREs) of the 3′ untranslated region of target mRNA [119]. An in vitro RNA binding experiment has shown that the CCCH zinc-finger protein OsTZF1 in rice binds to the 3’ untranslated region of mRNA, and OsTZF1 only binds to U homopolymers with the RNA gel electrophoresis mobility shift assay (REMSA) [120]. There are roughly three types of AREs [121]. Type I contains several scattered AUUUA motifs in U-rich regions [122]. Class II contains at least two repeated UUAUUUA (U/A) (U/A) sequences [123]. Class III belongs to the U-rich region, without typical features and the AUUUA motif [124]. Further in vivo experiments in plants have found that AtTZF1 can cause the degradation of ARE-containing mRNA [125]. Research shows that ribonucleoprotein domains or RNA binding domains (RRMs) are present in many CCCH zinc-finger proteins, indicating that these proteins can recognize and bind to RNA [126], and the RRM consensus sequence is Lys⁄Arg-Gly-Phe⁄Tyr-Gly⁄Ala-Phe⁄Tyr-Val⁄Ile⁄Leu-X-Phe⁄Tyr or Ile⁄Val⁄Leu-Phe⁄Tyr-Ile⁄Val⁄Leu-X-Asn-Leu [126]. By contrast, no plant CCCH zinc-finger proteins have the specific amino acids required to specifically recognize the ARE sequence, and the sequences are expected to be rich in U or G [127]. Wang et al. [63] used the three-dimensional structure of the TIS11D CCCH zinc finger as a template [128] to predict the conformation of the Arabidopsis zinc-finger protein AtC3H14 bound to RNA. Each zinc finger has a KTEL(V) residue at the N terminus, which forms a key interface for RNA binding. Each of the two zinc fingers forms a pocket for RNA, which accommodates the two nucleotide residues U6 and U2 of RNA [63]. In further research, the plant-specific TZF motif (RR-TZF) and RR sequence in AtTZF1 are necessary for the protein’s RNA-binding function, and transient expression analysis of plant protoplasts has further confirmed that AtTZF1 can bind to AREs and lead to mRNA degradation [125]. However, the researchers in that study did not use a natural mRNA target, but rather a generic ARE for research, so we lack direct evidence that plant CCCH zinc-finger proteins function in post-transcriptional regulation of gene expression in vivo. Recent research has found that two putative mRNA binding domains, LOTUS/OST-HTH and RRM, were detected in the AtC3H18L protein sequence; this CCCH zinc finger protein has been implicated in stop codon read-through in Arabidopsis [98].

CCCH proteins can regulate gene expression by affecting RNA metabolic processes, including degradation, cleavage, export, and polyadenylation [103]. The RNA metabolic functions of CCCH zinc-finger proteins affect plant development, growth, and abiotic stress responses [118]. In Arabidopsis, the RNA-binding protein HUA1 participates in flower development by regulating the pre-mRNA processing of *AGAMOUS* [129,130,131]. FRIGIDA-ESSENTIAL 1 (FES1) may promote the winter annual growth habits of Arabidopsis by affecting the mRNA levels of *FLOWERING LOCUS C* (*FLC*) in a FRIGIDA-dependent manner [132]. The RNA-binding CCCH zinc-finger protein AtCPSF30 also interacts with calmodulin, but in the presence of calmodulin, its RNA binding activity is reduced; in addition, AtCPSF30 interacts with itself, which may promote RNA processing [133]. Recombinant AtTZF3 and AtTZF2 have RNase activity in vitro, indicating that they may participate in mRNA processing [109]. AtTZF1 may be involved in regulating downstream genes at the post-transcriptional level to integrate plant growth and abiotic stress signals [114]. KHZ1 and KHZ2 may regulate flowering and senescence at the post-transcriptional level [102]. Further studies have found that KHZ1 and KHZ2 can inhibit the splicing efficiency of FLC pre-mRNA and promote flowering through autonomous pathways in Arabidopsis [134].

## 5. Interactions of CCCH Zinc-Finger Proteins with Other Proteins

CCCH zinc-finger proteins can also exert their functions by interacting with other proteins [135]. The mammalian CCCH zinc-finger protein TTP interacts with many different types of proteins through its N terminus, C terminus, or zinc-finger structure [136]. The mammalian CCCH zinc finger-protein ZAP interacts with the virus Nsp9 protein, thereby inhibiting the replication of the Porcine Reproductive and Respiratory Syndrome virus, and its interaction position is positioned to the zinc-finger domain of ZAP [135]. Because mammalian CCCH zinc finger-proteins and plant CCCH zinc-finger proteins have few homologous sequences except for the zinc-finger domain, protein interactions mediated through the N terminus or C terminus of the protein are not conserved in plant CCCH zinc-finger proteins. Plant CCCH-type zinc-finger proteins mainly interact with other proteins through their zinc-finger domains [84]. In cotton, a yeast two-hybrid screen showed that GhZFP1 mediates plant tolerance through interaction with GZIRD21A and GZIPR5 [84]. Further, these proteins that bind to the CCCH zinc-finger protein in the cytoplasm can shuttle to the nucleus and may activate the plant’s signaling pathway under stress conditions [84]. In Arabidopsis, AtTZF4, AtTZF5, and AtTZF6 interact with MARD1 and RD21A proteins through their zinc-finger motifs [97]. On the other hand, CCCH zinc-finger protein interaction may mediate post-transcriptional regulation in plant development responses [97].

## 6. CCCH Zinc-Finger Proteins and Plant Hormones

Plant CCCH zinc-finger proteins are effective regulators of hormone-mediated stress responses [91,137]. Many CCCH-type zinc-finger proteins are induced by ABA, gibberellin (GA), and jasmonic acid (JA), and these CCCH-type zinc-finger proteins play important roles in various hormone-mediated signaling pathways [120,138].

In Arabidopsis, the CCCH zinc-finger protein SOMNUS is involved in a signaling pathway mediated by ABA and GA. A loss-of-function *som* Arabidopsis mutant line has a lower content of endogenous ABA and a higher content of endogenous GA. In vivo experiments found that *SOMNUS* is directly activated by PIL5, indicating that the SOMNUS regulates ABA and GA metabolism downstream of *PIL5*. SOMNUS is also involved in plant pigment-mediated signaling pathways, regulating the expression of genes related to hormone metabolism downstream of PIL5 during seed germination [139]. 

Overexpression lines for *AtTZF1*, *AtTZF4*, *AtTZF5*, and *AtTZF6* exhibit a phenotype characteristic of enhanced ABA function and reduced GA function. Loss-of-function mutants for these genes are more sensitive to high salt, cold temperature, and drought stresses [73,140]. AtTZF2 and AtTZF3 also participate in the ABA signaling pathway. Expression of *AtTZF2* and *AtTZF3* is significantly induced under ABA treatment. *AtTZF2* and *AtTZF3* Arabidopsis overexpression lines have enhanced tolerance to high salt, osmotic, and ROS stresses. In contrast, the silencing lines for these genes have reduced salt and drought tolerance [105,106,109]. In rice, the ABA-induced CCCH zinc-finger protein OsC3H47 reduces sensitivity to ABA and enhances drought tolerance [141].

JA-mediated signaling pathways may also involve CCCH zinc-finger proteins [72,120,142]. JA can control leaf senescence in plants through the production of ROS [143]. GhTZF1 from cotton, OsTZF1 and OsTZF2 from rice, and AtTZF2 and AtTZF3 from Arabidopsis are all involved in JA-mediated leaf senescence [72,106,120,142]. AtTZF2 and AtTZF3 both participate in plant drought tolerance and growth through ABA and JA signaling pathways. Gene chip analysis and RT-qPCR assays have shown that the differentially expressed genes in Arabidopsis *Attzf2* and *Attzf3* overexpression lines are involved in JA, ABA, and biotic and abiotic stress processes [109]. Several JA biosynthesis genes and JA response genes are significantly upregulated in *OsTZF2* gene silencing lines, while the same genes are downregulated in *OsTZF2* overexpression lines, indicating that OsTZF2 negatively regulates the JA signaling pathway [72]. 

## 7. The Roles of CCCH Zinc-Finger Proteins in Abiotic Stress

CCCH-type zinc-finger proteins widely participate in plant responses to abiotic stresses such as oxidative, salinity, drought, flooding and cold temperature stresses. 

### 7.1. Oxidative Stress

Many abiotic stresses induce the production of reactive oxygen species (ROS) in plants [144]. Excessive accumulation of ROS will produce oxidative stress, leading to cell death and even plant death [145]. Mitochondria, chloroplasts, and peroxisomes in plant cells are the main production sites of ROS. In the face of abiotic stress, plants have a replication mechanism to regulate the balance of reactive oxygen species, which mainly include enzymatic clearance mechanisms and non-enzymatic clearance mechanisms [146]. The CCCH zinc-finger protein is also involved in this process.

Hydrogen peroxide, abscisic acid and salt stress can significantly induce the expression of AtOZF1. Compared with the wild type, the Arabidopsis *atozf1* mutant lines have reduced resistance to oxidative stress, while the overexpression lines improves the antioxidant capacity of the plant. Compared with the wild type, the *atozf1* mutant has accumulated more MDA. In addition, the activity of CAT and guaiacol POD in *atozf1* mutant is lower, and the expression of antioxidant genes such as *APX1* and *AtGSTU5* is down-regulated under oxidative stress. Experimental results show that AtOZF1 plays an important role in the resistance of Arabidopsis to oxidative stress [105].

### 7.2. Salt Stress

More than 8 × 10^8^ hm^2^ of land worldwide is affected by soil salinization, and this problem continues to worsen [147,148,149,150]. Soil salinization severely affects seed germination, crop growth, and productivity, and is a major factor limiting global agricultural production [151,152,153,154]. NaCl is the main component causing salt stress [155,156,157,158]. High concentrations of salt in the soil reduce the water potential at the root surface, which affects water absorption by the roots and reduces the water use efficiency of plants, leading to osmotic stress and ionic stress, which can generate oxidative stress [159,160,161,162]. 

Some CCCH-type zinc-finger proteins are induced by salt stress and are closely associated with salt stress tolerance in plants [21]. For example, in rice subjected to various salt treatments, *OsC3H33*, *OsC3H37*, and *OsC3H50* are all induced [163], and NaCl induces the expression of GhZFP1 in cotton, indicating that these CCCH zinc-finger proteins may function in regulating plant salt stress tolerance [84]. AtSZF1 and AtSZF2 in Arabidopsis [101]; OSC3H33, OSC3H37, OSC3H47 and OSC3H50 in rice [141,163]; GhTZF1 in cotton [142] all regulate the adaptability of plants to salt stress. 

CCCH zinc-finger proteins participate in salt tolerance by various mechanisms. One way these proteins improve salt tolerance in plants is by maintaining ion homeostasis. Overexpression of *GhZFP1* in transgenic tobacco plants significantly increases their salt tolerance by affecting Na^+^ homeostasis and K^+^ acquisition [84]. *AtZFP1* improves salt tolerance in Arabidopsis by increasing K^+^ content and decreasing the Na^+^/K^+^ ratio [21]. 

CCCH zinc-finger proteins also improve the salt tolerance of plants by controlling genes related to salt stress. AtSZF1 and AtSZF2 participate in the response of Arabidopsis to salt stress by negatively regulating the expression of salt stress response genes. The single Arabidopsis mutants *atszf1-1* and *atszf2-1* show a significant increase in the expression of salt stress-related genes, including *KIN1*, *RD29A*, *COR15A*, and *COR47*. The Arabidopsis double mutants have significantly reduced salt tolerance, while the overexpression lines for these two genes have significantly improved salt tolerance [101]. Later research has transformed *AtSZF2* into soybeans. Under high salt stress, it has been found that the ectopic expression of *AtSZF2* gene could significantly reduce the ion leakage and increase the chlorophyll content of transgenic lines. The detection showed that *AtSZF2* gene could regulate the expression of ABA/stress response genes in transgenic soybeans, making them salt tolerant. AtZFP1 improves the salt tolerance of plants by activating the downstream genes *SOS1*, *AtGSTU5*, and *AtP5CS1* [21].

Post-transcriptional regulation is another mechanism by which CCCH zinc-finger proteins improve the salt tolerance of plants. OsTZF1 confers stress tolerance in rice: *OsTZF1* overexpression lines show significantly higher tolerance to salt stress than the control and the *OsTZF1* silenced lines. OsTZF1 has no transcriptional activation activity, so it does not regulate plant salt tolerance at the transcriptional level [120]. RNA electrophoretic mobility shift assays have shown that OsTZF1 can bind U-rich and ARE motifs of mRNAs, and microarray analysis has shown that OsTZF1 may post-transcriptionally regulate the transcript levels of salt stress-related genes, including *Tic32*, *RNS4*, and *WRKY45* [120].

CCCH zinc-finger proteins can also increase the ability of plants to scavenge reactive oxygen species (ROS). Under high salt stress, OsTZF1 can reduce ROS damage by regulating genes related to redox homeostasis, such as those coding for ferritin and metallothionin, and genes encoding antioxidant enzymes, such as peroxidase (POD) and glutathione S-transferase (GST) [120]. The overexpression of *BoC3H* improves the salt tolerance of transgenic *Brassica oleracea* [164]. BoC3H may improve salt tolerance in broccoli by reducing relative conductivity, hydrogen peroxide (H_2_O_2_), and malondialdehyde (MDA), and by increasing the levels of catalase (CAT), POD, and superoxide dismutase (SOD).

CCCH zinc-finger proteins also help plants adjust to high salt environments through the ABA signaling pathway. AtOZF2 overexpression lines significantly improve plant salt resistance, while the antisense *atozf2* Arabidopsis line significantly reduces plant salt resistance. Further studies have found that AtOZF2 participates in plant salt resistance through a signaling pathway mediated by ABA insensitive 2 (ABI2) [106]. Under salt stress, the stress response genes *RAB18*, *COR15A*, and *RD22* in the ABA signaling pathway are significantly upregulated in an *AtC3H17* overexpression Arabidopsis line compared with the control. These results indicate that AtC3H17 may regulate plant salt stress responses through an ABA-dependent pathway [95].

### 7.3. Drought Stress

Under drought conditions, plant root activity decreases, respiration decreases, and root absorption and the transport of water and mineral elements is inhibited [6,165,166,167]. At the same time, leaf growth, leaf area, stomatal index, the opening and closing of stomata, and chlorophyll content decreased, seriously disrupting photosynthesis and respiration [18,168,169,170]. Under drought stress, the ROS levels in plants increase, followed by membrane lipid delipidization or oxidation to form MDA [171,172,173,174]. Drought stress induces the expression of many CCCH zinc-finger protein genes. Through bioinformatics and phylogenetic analysis, 68 CCCH genes in 7 subfamilies have been identified in maize. Among them, 12 CCCH zinc-finger protein genes were found to have putative stress-responsive cis-elements in their promoter regions [64]. Real-time quantitative polymerase chain reaction (RT-qPCR) analysis has found that these genes have five different expression patterns among different tissues and are significantly induced by ABA and drought treatment [64]. In *Aegilops tauschii* Coss., 36 CCCH zinc-finger family genes have been identified by comprehensive computational analysis, among which *AetTZF1* shows the strongest expression under drought stress [67]. Drought and ABA could significantly induce the expression of *OsC3H10* in rice [104].

CCCH zinc-finger proteins increase the drought tolerance of plants in a variety of ways. One way is by regulating the function of stomata. *AtTZF1* overexpression lines of Arabidopsis are significantly more drought-resistant than the wild type. These drought-tolerant transgenic lines have abnormal stomatal closure and lower stomatal conductance [140]. PeC3H74 is a CCCH zinc-finger protein located on the plasma membrane of *Phyllostachys edulis* (Carriere) J. Houzeau with self-activating activity. Under drought treatment, the stomata closure rate of the PeC3H74 transgenic Arabidopsis line has been found to be significantly higher than that of the wild type [175].

CCCH zinc-finger proteins directly regulate downstream genes related to drought stress to enhance plant drought tolerance at the transcription level. *AetTZF1* transgenic lines of Arabidopsis have a higher germination rate and a stronger root system than the wild type under drought stress. AetTZF1 improves the drought tolerance of the transgenic plants by increasing the expression level of drought stress-related genes, such as *CBF1*, *CBF2*, *DREB2A*, and *COR47* [67]. It has been found that the root specific overexpression of *OsC3H10* is not enough to induce drought resistance in rice, and the overexpression of *OsC3H10* in the whole plant enhances the drought resistance of rice. Transcriptome analysis has shown that *OsC3H10* overexpression lines could increase the expression level of stress response related genes, including LATE EMBRYOGENESIS ABUNDANT PROTEINs (LEAs), PATHOGENESIS RELATED GENEs (PRs) and GERMIN-LIKE PROTEINs (GLPs) [104].

CCCH zinc-finger proteins directly regulate downstream genes related to drought stress to enhance plant drought tolerance at the post-transcription level. The expression of Oryza sativa CCCH-tandem zinc finger protein 5 (OsTZF5) can be induced by drought stress [176]. Under drought, OsTZF5 overexpressing lines have shown an improved survival rate and growth retardation. When under the control of the stress-inducible OsNAC6 promoter, OsTZF5 rice overexpressing lines have increased the survival rate of rice under drought stress without growth retardation. The RNA electrophoretic mobility shift assay (REMSA) has shown that OsTZF5 can bind to the 3′ untranslated region of RNA in vitro, which indicates that OsTZF5 may confer plant drought tolerance by regulating RNA. Microarray analysis has shown that compared with the wild type, the OsTZF5 overexpression line has 609 up-regulated genes and 196 down-regulated genes, respectively. Representative genes up-regulated are *PR1*, *salT* and *GolS2*, and representative genes down-regulated are *OsSAM3* and *flavin monooxygenase* [176].

CCCH zinc-finger proteins also enhance drought tolerance through a signaling pathway mediated by ABA. *AtTZF2* and *AtTZF3* overexpression lines are highly sensitive to ABA and have reduced transpiration, increased drought tolerance, and altered expression of genes related to ABA and abiotic stress processes [109]. Drought stress mimicked by the osmotic stress induced by a 10-day polyethylene glycol (PEG) treatment has been found to significantly inhibit the growth of *OsC3H47* overexpressing lines and wild-type rice, and both genotypes showed accelerated leaf senescence and leaf curling. After recovery, the survival rates and PEG-induced drought tolerance of *OsC3H47* overexpression lines were significantly higher than those of the control. The CCCH zinc-finger protein OsC3H47 reduces the sensitivity of rice to ABA and improves the drought tolerance of rice. OsC3H47 is located downstream of the ABA signaling pathway and plays an important role in post-transcriptional regulation [141]. PeC3H74 transgenic Arabidopsis seedlings have better root growth than that of the wild type under 10 μM ABA treatment. This indicate that PeC3H74 may enhance the drought tolerance of plants through ABA-dependent signaling pathways [175].

CCCH zinc-finger proteins also enhance drought tolerance through improving ROS scavenging capacity. For example, transformation of Arabidopsis plants with *GhTZF1* reduces drought damage and delays leaf senescence by increasing the content of SOD and POD [142]. Compared with the wild type, PeC3H74 transgenic Arabidopsis has significantly improved drought tolerance, including the accumulation of less H_2_O_2_ content, less electrolyte leakage, and MDA content [175].

### 7.4. Flooding Stress

Over the past few decades, droughts and floods have become more frequent, and their impact on the growth and development of crops has become more serious [177,178,179]. Flooding stress significantly inhibits the growth and development of crops, significantly reduces yield and quality, and can kill plants in severe cases [180,181].

CCCH zinc-finger proteins participate in the response to flooding stress in different ways. In rice, hypoxia stress induces the expression of the CCCH zinc-finger genes *OsCCCH*-*Zn*-*1*, *OsCCCH*-*Zn*-*2*, and *OsCCCH*-*Zn*-*3*. The expression levels of *OsCCCH-Zn-1* also increase significantly during flooding and in response to ABA. However, this gene is not induced by H_2_O_2_, cold stress, or salt stress, indicating that OsCCCH Zn-1 may exclusively regulate responses to flooding stress through the ABA signaling pathway, and rarely participates in responses to other abiotic stresses [182]. The detailed mechanisms of how CCCH zinc-finger proteins function in plant adaptation to flooding need further investigation.

### 7.5. Cold Stress

Low temperature is one of the abiotic stresses most likely to be encountered during the plant life cycle, and it greatly limits the geographical distribution, growth and development, yield quality, and post-harvest viability of crops [183,184,185,186]. Understanding the physiological and biochemical effects of low temperatures on plants and the molecular mechanisms of plant responses to low temperature stress is essential for breeding crop varieties with improved tolerance to low temperatures [187,188,189,190].

CCCH zinc-finger proteins improve the cold tolerance of plants by directly regulating the expression of downstream cold-related genes as transcription factors. For example, Arabidopsis plants overexpressing *AtTZF1* are superior to the wild type in cold tolerance. In one study, under low temperature conditions, the survival rate of AtTZF1 overexpression lines was significantly higher than that of the control. The cold-tolerant phenotype of the Arabidopsis lines were consistent with the upregulated expression of the ABA/low temperature-related genes *KIN1*, *RD29A*, and *COR15A* [140]. DgC3H1 is a nuclear-localized CCCH zinc-finger protein isolated and cloned from *Chrysanthemum morifolium* Ramat. In another study, the overexpression line improved the low temperature tolerance of *Chrysanthemum*. The DgC3H1 antisense expression lines reduced the low temperature tolerance of *Chrysanthemum*. Under low temperature stress, the content of proline and soluble sugar in the DgC3H1 overexpression lines increased, and the activity of POD and SOD increased. In addition, compared with the wild type, low temperature stress-related genes such as *DgCOR413*, *DgDREBa*, *DgCSD1* and *DgCSD2* were up-regulated in the overexpression line and down-regulated in the antisense line. The results showed that DgC3H1 can act as a transcription factor to improve the resistance of *Chrysanthemum* to cold stress [191].

CCCH zinc-finger proteins also enhance cold tolerance via ABA signaling pathways. In switchgrass (*Panicum virgatum* L.), low temperature treatment can significantly induce the expression of the CCCH zinc-finger protein PvC3H72. *PvC3H72* overexpression transgenic switchgrass has significantly improved cold tolerance compared to wild-type plants at 4 °C. In a study, at −5 °C, the transgenic line had a higher survival rate than the control, as well as a higher relative water content and more stable cell membranes. PvC3H72 improves the cold tolerance of switchgrass transgenic plants by regulating the expression of the ICE1-CBF-COR complex and ABA signaling pathway genes [192]. 

### 7.6. Multiple Stresses

Some CCCH zinc-finger proteins also participate in the responses to multiple abiotic stresses. In rice, OsTZF1 can significantly improve the ability of plants to resist salt, drought, and oxidative stress. Significant up-regulation of many biotic and abiotic stress-related genes in *OsTZF1* overexpression lines, including disease-related proteins, transcription factors, peroxidase, dehydrin, metal detoxification protein, and ROS scavenging genes has been detected. Further studies have found that as the level of ROS in cells decreases, the expression of genes related to stress also decreases; therefore, OsTZF1 may confer tolerance to abiotic stresses in rice by enhancing tolerance to oxidative stress. In addition, as OsTZF1 can bind to mRNA in vitro, it is speculated that OsTZF1 may regulate abiotic stress through RNA metabolism [120]. In *Ipomoea batatas* (Linn.) Lamarck, *IbC3H18* was cloned and analyzed, and the *IbC3H18* overexpression transgenic line was found to have significantly enhanced drought tolerance, salt tolerance, and antioxidant capacity, while an *IbC3H18* RNA interference silencing line had significantly reduced tolerance. IbC3H18 can directly bind to the promoter regions of the salt tolerance-related gene *SOS5*, the active oxygen scavenging gene *CCS*, and the ABA receptor gene *PYL8* to regulate their expression. Based on the results of RNA sequencing and on physiological and biochemical indicators, overexpression of *IbC3H18* increases photosynthetic activity by activating the ROS scavenging system and ABA signaling pathway, and maintains the ionic–osmotic balance to enhance the stress tolerance of sweet potatoes [81].

## 8. Mechanisms That Regulate CCCH Zinc-Finger Proteins under Abiotic Stresses

Plants rapidly respond to a specific abiotic stress based on which type of sensor is stimulated, as this determines which series of responses is activated [193,194]. For instance, the sphingolipid glycosyl inositol phosphorylceramide (GIPC) is involved in sensing salt stress [195], REDUCED HYPEROSMOLALITY INDUCED CALCIUM INCREASE 1 (OSCA1) is an osmotic stress sensor [196], and CHILLING TOLERANCE DIVERGENCE 1 (COLD1) is a cold stress receptor [197,198]. Upon sensing the stress signal, plants produce second messengers (such as a Ca^2+^ pulse) in the cytoplasm, which act on a series of downstream genes, including CCCH zinc-finger protein genes. However, it is unclear how the upstream stress signals are accurately transmitted to the CCCH zinc finger proteins. The CCCH zinc-finger proteins then regulate plant development and abiotic stress by modulating gene expression, MAPK-mediated protein phosphorylation, and ABA signaling at the transcriptional and post-transcriptional levels via different signaling pathways.

Some CCCH zinc-finger proteins function as transcription factors that control various genes at the transcriptional level under abiotic stress [71,192]. CCCH zinc-finger proteins such as PEI1, HUA1, OsDOS, SOMNUS, AtSZF1, AtZFP1, SAW1 and OsC3H10 are located in the nucleus, where they act as transcription factors [13,103,104]. Transcription factors bind to the promoters of stress-related genes as part of a signaling pathway that transmits relevant stress stimuli, initiates the expression of these genes, and helps the plant adapt to stress by producing particular proteins or metabolites [199]. For example, AtSZF1 [101] and AtZFP1 [21] improve the salt tolerance of plants as transcription factors.

CCCH zinc-finger proteins also function at the post-transcriptional level by modulating RNA metabolism in plants subjected to stress. CCCH zinc-finger proteins localize to PBs and SGs, which can combine with specific RNA elements to initiate RNA degradation in PBs and regulate plant development, growth, and stress responses in SGs [140]. Furthermore, AtTZF1, AtTZF4, AtTZF5, AtTZF6, OsTZF1 and OsC3H10 proteins, which shuttle between the cytoplasm and the nucleus, co-localize with PBs and SGs [104,120,140]. For example, AtTZF1 [140] and OsTZF1 [120] improve plant resistance to abiotic stresses by regulating stress-related genes, possibly via RNA metabolism. Although AtTZF1 and OsTZF1 can bind to specific RNA elements, there is currently no evidence that this combination leads to mRNA degradation [114].

Moreover, CCCH zinc-finger proteins improve plant stress tolerance through the ABA signaling pathway. Many CCCH zinc-finger proteins, such as AtTZF1, AtTZF2, AtTZF3, AtTZF4, AtTZF5, AtTZF6, OsC3H47 and OsTZF5, are upregulated in response to ABA [140,141,176]. For example, AtTZF10/11 [101] and AtC3H17 [95] may regulate the adaptability of plants to high-salt conditions through an ABA-dependent signaling pathway. These experimental data indicate that CCCH zinc-finger proteins extensively improve plant stress resistance through ABA signaling.

In addition, CCCH zinc-finger proteins such as GhZFP1 [84] may improve plant stress tolerance through protein interaction. However, GhZFP1 may play different roles in abiotic stress, potentially acting as an RNA binding protein to regulate the RNA metabolism of target genes induced by salt stress [84].

In summary, when plant receptors sense a stress signal, the signal is amplified by secondary messengers and transmitted to CCCH zinc-finger proteins in some unclear way, which improves the stress tolerance of the plant in different ways at the transcriptional and post-transcriptional levels and through protein–protein interactions. A putative regulatory mechanism by which plant CCCH zinc-finger proteins mediate plant abiotic stress responses is shown in Figure 1. In the nucleus, CCCH zinc-finger proteins function as transcription factors that bind to specific *cis*-elements in the promoters and activate or repress the expression of stress-related genes, improving plant stress resistance through ABA signaling pathways (a). In the cytoplasm, CCCH zinc-finger proteins bind to target mRNAs at U-rich sequences and regulate RNA metabolism in PBs and SGs (b). In the cytoplasm, CCCH zinc-finger proteins can also interact with stress-related proteins, such as MARD1, RD21A, and PR5, to improve the plant’s resistance (c). In addition, CCCH zinc-finger proteins such as ZFPs interact with PBs and SGs shuttling between the cytoplasm and nucleus to improve the plant’s abiotic stress resistance (d). Finally, the gene IDs of CCCH zinc-finger proteins, CCCH motifs, functions related to stress, mode of action, and related references are summarized in Appendix A.

## 9. Conclusions and Perspectives

Despite our progress in understanding how CCCH zinc-finger proteins contribute to plant responses to abiotic stress, more specific receptors for sensing abiotic stresses, such as salt, osmotic, and low temperature stresses, remain to be identified. Emerging technologies, such as genome-wide association analysis, CRISPR–Cas9 genome editing, and single-cell sequencing will be very helpful for examining the mechanisms by which different CCCH zinc-finger proteins interact with various sensors to improve the abiotic stress tolerance of plants.

More detailed studies are needed to unravel how secondary messengers such as calcium pulses induced by abiotic stress initiate CCCH zinc-finger protein–mediated regulation of post-transcription, protein interaction, and shuttling between the cytoplasm and nucleus, although Ca^2+^-mediated ABA or MAPKs transcription regulation is clear.

Plant CCCH zinc-finger proteins can be located on PBs and SGs, and bind to specific RNA elements to initiate RNA degradation. However, no natural RNA binding target site has been found on plant CCCH zinc-finger proteins. Therefore, it is necessary to seek the natural RNA binding site of plant CCCH zinc-finger proteins through in vivo experiments. RNA immunoprecipitation (RIP) and cross-linking and immunoprecipitation (CLIP) may be efficient methods to determine the function of plant CCCH zinc-finger proteins in post-transcriptional regulation [75,200].

Most research on plant CCCH zinc-finger proteins has focused on model plants such as Arabidopsis and rice, which are highly sensitive to abiotic stress. Analyzing the sensors, signal transduction, and molecular regulatory networks of CCCH zinc-finger proteins in plants that are extremely tolerant to abiotic stress, such as heavy metals or arsenic hyperaccumulator plants, halophytes and xerophytes, would provide valuable insight into the mechanism by which CCCH zinc-finger proteins contribute to abiotic stress tolerance. Although individual studies have found that the CCCH zinc-finger protein regulates proteins related to heavy metal detoxification [120], specific studies on CCCH zinc-finger proteins’ regulation of plant heavy metal (Cd, Cr, Al, Cu, Zn, etc.) or arsenic stress have not been reported, which shows that there is still a large research space between CCCH zinc-finger proteins and plant heavy metal stress, which is our future research direction.

## Figures and Tables

**Figure 1 ijms-22-08327-f001:**
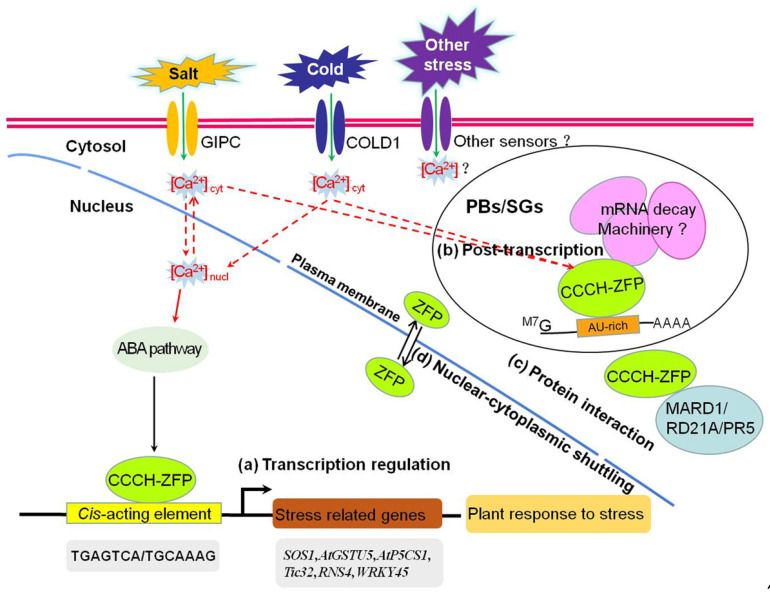
Putative regulatory mechanism by which CCCH zinc-finger proteins mediate abiotic stress tolerance in plants. (**a**) Transcription regulation of CCCH zinc-finger protein in the nucleus under abiotic stress. (**b**) Post-transcriptional regulation of CCCH zinc-finger protein in processing bodies (PBs) and stress granules (SGs) of the cytoplasm under abiotic stress. (**c**) Protein interaction of CCCH zinc-finger protein in the cytoplasm under abiotic stress. (**d**) Shuttling of CCCH zinc-finger protein between the nucleus and cytoplasm under abiotic stress. Solid arrows indicate processes that have been verified, and dotted arrows indicate processes that need to be clarified.

## Data Availability

Data is presented in manuscript and in Appendix A.

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
