# Peer review of "The Roles of CCCH Zinc-Finger Proteins in Plant Abiotic Stress Tolerance"

_ijms, 2021, doi:10.3390/ijms22158327_

Round 1

Reviewer 1 Report

Review of the manuscript: ID ijms-1285258

Title: “  Review  The Roles of CCCH Zinc-Finger Proteins in Plant Abiotic Stress  Tolerance”

The manuscript constitutes an extensive review of the role of CCCH Zinc-Finger Proteins in relation to plants. It regards their structure, their functions in various plants especially their role in plant stress response and interaction with hormones, their cellular localization, and interaction with other proteins and with mRNA. The bibliography utilized is large and appropriate, just one case in my opinion is out of context, which is reference 10.

There are two main drawbacks to this review. The first is that there is no mention, neither in the introduction where the authors amply mention halophyte plants as examples of plants living in harsh conditions, of the heavy metals land arsenic hyperaccumulator plants. Such plants live in extreme conditions too, thus they should be mentioned in the introduction. Moreover, there should be a subparagraph in Paragraph 6 (The Roles of CCCH Zinc-Finger Proteins in Abiotic Stress) dedicated to heavy metal stress (Cd, Cr, Al, Cu, Zn, etcetera) and the function of the zinc finger proteins in plant survival in conditions of heavy metals pollution and how and if Zink finger proteins play any role in the metals (and arsenic) hyperaccumulator habit. In addition, in paragraph 6 the subparagraph on Oxidative stress should be moved at the top, because in many stresses, starting from salt and drought the authors mention ROS and oxidative metabolism. For the same reason, that the hormones are frequently mentioned in all the stresses subparagraph I would move paragraph 7 on hormones interaction with Zinc finger proteins before paragraph 6 on stresses. Finally, especially in the paragraph on hormones, several biotic stresses have been mentioned (see my annotations in the text). I do not know if they are appropriate in this context of abiotic stress and I would suggest finding an alternative abiotic stress comparison and eliminating all the references to biotic stress and parasites –plant interaction.

All the text has been annotated for minor comments and is attached as a pdf file. Please read it thoroughly because there are important points to be considered by the Authors also among the many annotations.

Author Response

Response to Reviewer 1 Comments

Dear Editor and Reviewers,
Thank you very much for your suggestions and critical comments on our paper submitted to International Journal of Molecular Sciences with a revised title of “The Roles of CCCH Zinc-Finger Proteins in Plant Abiotic Stress Tolerance”. We have revised the manuscript according to the comments and suggestions. The main revised parts are marked in red in the revised version. The following is a detailed explanation of how we have complied with the reviewers’ suggestions.

Point 1: The first is that there is no mention, neither in the introduction where the authors amply mention halophyte plants as examples of plants living in harsh conditions, of the heavy metals land arsenic hyperaccumulator plants. Such plants live in extreme conditions too, thus they should be mentioned in the introduction. 
Response 1: We agree with you, metals hypeaccumulators plants also live in extreme enviroment, and play improtant roles in adaptting to adversity. So we added related content about arsenic hyperaccumulator plants in the Introduction section. The details are provided in lines 43-46 of the revised version.

Point 2:
In the Introduction section: Throughout their lives, plants are exposed to various complex environmental conditions [1-4], in the references [1-4] there are no mentions of metals hypeaccumulators plants which live in extreme enviroment.
Response 2: We agree with you. We added relevant reference in this section. In the Introduction section, throughout their lives, plants are exposed to various complex environmental conditions [1-5]. The details are provided in line 32 of the revised version.
5. Rascio, N.; Navari-Izzo, F., Heavy metal hyperaccumulating plants: how and why do they do it? And what makes them so interesting? Plant science : an international journal of experimental plant biology 2011, 180, (2), 169-81. 

Point 3: ‘ROS’ in the Abstract section should be described as oxidative stress, in line 17.
Response 3: Done. The details are provided in line 20 of the revised version.

Point 4: and excell of metals, in line 31.
Response 4: Done. We supplemented heavy metal stress and added relevant references. The details are provided in line 35 of the revised version.
11. Zubair, M.; Khan, Q. U.; Mirza, N.; Sarwar, R.; Khan, A. A.; Baloch, M. S.; Fahad, S.; Shah, A. N., Physiological response of spinach to toxic heavy metal stress. Environmental science and pollution research international 2019, 26, (31), 31667-31674.

Point 5: REF.10 is out of scope  in the whole manuscript, plese delete it, in line 32. 
Response 5: Done. We have delete the REF.10. and the details are provided in line 35 of the revised version.

Point 6: Insert botanical authority of the plants. Please do it in all the manuscript the first time you mention the scientific name of a plant.
Response 6: Done. The details are provided in lines 58-62, 65, 69-70, 362, 371, 444 and 454.

Point 7: However, in the last part of the manuscript there is ample mentioning also of biotic stress, please mention it here.
Response 7: Done. In the revised version, we deleted the content related to biotic stress, and the paper still focuses on the role of CCCH zinc-finger proteins in plant responses to abiotic stress.

Point 8: Many TZF proteins have an plant unique…in line 94.
Response 8: Done. Many TZF proteins have a plant unique…The details are provided in line 101 of the revised version.

Point 9: ‘The domain structures of different motifs in CCCH proteins described in more detail below, where we discuss protein localization and function.’ This sentence in line 110-111 is grammatically incorrect, please recast it.
Response 9: Done. We recasted the sentence .The different domain structures of CCCH proteins will be described more details in the below section, where we discuss the protein localization and function. The details are provided in lines 116-118 of the revised version.

Point 10: Studies of CCCH zinc-finger proteins have provided some information about their subcellular locations. This sentence in 1ine 114-115 is a repetition, cut.
Response 10: Done. We have delete this sentence. The details are provided in line 121 of the revised version.

Point 11: Clarify the related proteins we mentioned above in line 127.
Response 11: Done. NESs are present in 54 CCCH proteins in Arabidopsis, including the whole 11 proteins of IX subfamily, and the proteins such as AtTZF1, AtTZF2, AtTZF3, AtTZF4, AtTZF5, AtTZF6, AtTZF7 and AtTZF11 mentioned above were the representative members. The details are provided in lines 131-134 of the revised version.

Point 12: Insert the number of this references. Here there is a bit of confusion because wang 2020 are ref. number 34, 93,96,169,185. It is important that you insert the correct number
Response 12: Done. The details are provided in lines 150, 156 and 209 of the revised version.

Point 13: Low case letter: ‘the’ in line 148.
Response 13: Done. The details are provided in line 154 of the revised version.

Point 14:Start the sentence with: Whilst, ILA1-interacrting protein... in line 168.
Response 14: Done. The details are provided in line 173 of the revised version.

Point 15:  In plant, Substitute with"In Arabidopsis" in line 207.
Response 15: Done. The details are provided in line 211 of the revised version.

Point 16:  cut "in arabidopsis" in line 208.
Response 16: Done. The details are provided in line 212 of the revised version.

Point 17: explain better the content of this sentence, as it is is a bit opaque. FRIGIDA-ESSENTIAL 1 (FES1) may promote the winter annual growth habit of Arabidopsis by affecting the FRIGIDA-mediated increase in FLOWERING LOCUS C (FLC) mRNA levels in line 209-211.
Response 17: Done. The details are provided in lines 212-215 of the revised version.

Point 18: Add "in Arabidopsis" at the end of the sentence in line 220.
Response 18: Done. The details are provided in line 223 of the revised version.

Point 19: this sentence is not clear, please rephrase it.  In line 222
Response 19: Done. CCCH zinc finger proteins also exert their functions by interacting with other proteins. The details are provided in lines 225-226 of the revised version.

Point 20: In this para I think it would be better to put the subpara on ROS first followed by all the others. 
Response 20: Done. The subpara of ROS stress was first introduced before other stress. The details are provided in lines 276-292 of the revised version.

Point 21: In addion I would insert a subpara on heavy metals (Cd, Cr, Al, Cu, Zn, etcetera) stress. The function of the zinc finger proteins in plant survival in conditions of heavy metals pollution and how and if Zink finger proteins play any role in the metals (and arsenic) hyperaccumulator habit.
Response 21: Done. Although individual studies have found that the CCCH zinc-finger protein regulates proteins related to heavy metal detoxification[192], we have not found a specific research about CCCH zinc-finger proteins in regulating plant heavy metal (Cd, Cr, Al, Cu, Zn, etcetera) stress, which shows that there is still a large research space between CCCH zinc finger proteins and plant heavy metal stress, which is also the research direction in the future, and we emphasize this in the Conclusions and Perspectives section. The details are provided in lines 560-568 of the revised version.
192. Jan, A.; Maruyama, K.; Todaka, D.; Kidokoro, S.; Abo, M.; Yoshimura, E.; Shinozaki, K.; Nakashima, K.; Yamaguchi-Shinozaki, K., OsTZF1, a CCCH-tandem zinc finger protein, confers delayed senescence and stress tolerance in rice by regulating stress-related genes. Plant Physiol 2013, 161, (3), 1202-16.

Point 22: which generate oxidative stress in line 250.
Response 22: Done. The details are provided in line 300 of the revised version.

Point 23: cut the semicolon in line 256.
Response 23: Done. The details are provided in line 306 of the revised version.

Point 24: The single Arabidopsis mutants... in line 266.
Response 24: Done. The details are provided in line 316 of the revised version.

Point 25: substitute this entence with: The arabidopsis double mutants have significantly...in line 267-268.
Response 25: Done. The details are provided in lines 317-318 of the revised version.

Point 26: insert the acronym GST in line 288.
Response 26: Done. The details are provided in line 337 of the revised version.

Point 27: the antisense atozf2 Arabidopsis line...in line 295.
Response 27: Done. The details are provided in line 343 of the revised version.

Point 28:Arabidopsis line...in line 299.
Response 28: Done. The details are provided in line 347 of the revised version.

Point 29: in what way? in line 306.
Response 29: Done. The details are provided in line 354 of the revised version.

Point 30: rice in line 344.
Response 30: Done. The details are provided in line 389 of the revised version.

Point 31: insert the acronym RNA EMSA in parenthses () in line 345.
Response 31: Done. The details are provided in line 391 of the revised version.

Point 32:Arabidopsis lines...in line 399.
Response 32: Done. The details are provided in lines 441-442 of the revised version.

Point 33: switchgrass transgenic plants...in line 416. 
Response 33: Done. The details are provided in line 459 of the revised version.

Point 34: this is a repetition, cut it… in line 422.
Response 34: Done. The details are provided in line 281 of the revised version.

Point 35:the Arabidopsis atozf1 mutant lines has reduced....in line 427.
Response 35: Done. The details are provided in line 286 of the revised version.

Point 36:in italics in line 443.
Response 36: Done. The details are provided in line 469 of the revised version.

Point 37: in which plant has this gene been cloned? In line 447.
Response 37: Done. The details are provided in line 471 of the revised version.

Point 38: since in all stresses subparagraph there is a mention to the interaction with some hormone , this para should be put before the para on abiotic stress
Response 38: Done. The details are provided in lines 242-273 of the revised version.

Point 39: som arabidopsis mutant line… in line 459.
Response 39: Done. The details are provided in line 248 of the revised version.

Point 40: Arabidopsis in line 469
Response 40: Done. The details are provided in line 258 of the revised version.

Point 41: Arabidopsis Attzf2 and Attzf3 in line 479.
Response 41: Done. The details are provided in line 269 of the revised version.

Point 42: which lines?  in line 482.
Response 42: Done. The details are provided in line 272 of the revised version.

Point 43: Here a biotic stress is mentioned, but the review is on abiotic stress, try to find some abioti alternative for this concept or delete it, in line 483-485.
Response 43: Done. We did not find a suitable alternative for abiotic stress, so we deleted it in the revised version.

Point 44: Again here a biotic stress is mentioned, try to find an alternative or delete it, in line 489-490.
Response 44: Done. We did not find a suitable alternative for abiotic stress, so we deleted it in the revised version.

Point 45: RNA degradation occurs in PBs, the other reactions in SGs, please specify, in line 518-519.
Response 45: Done. The details are provided in lines 504-505 of the revised version.

Point 46: Again here is the mention of a Biotic stress, find an abiotic alternative or eliminate it, in line 535.
Response 46: Done. We did not find a suitable alternative for abiotic stress, so we deleted it in the revised version.

Point 47: If this number (and the others (2), (3), (4) are referred to the Figure below, Figure 1, the Authors should mention here Figure 1 in the text…  mention it at the beginning of the paragraph before you start with the numbers. In anyway the paragraph is very confuse, I suggest to rewrite it more clearly making the reference to the figure 1 and its caption more direct.  in lines 548-555.
Response 47: Done. In order to prevent misunderstanding, in this paragraph, we firstly mention Figure 1, and then remake the reference to the Figure 1 as (a), (b), (c) and (d). The details are provided in lines 526-536 of the revised version.

Point 48:and also metals or Arsenic hyperaccumulator plants… in line 587.
Response 48: Done. The details are provided in lines 560-568 of the revised version.

For the Revised Main Manuscript, please see the attachment.

We have tried our best to improve the manuscript and have made substantial changes and deletions according to the reviewers’ comments. We earnestly appreciate the reviewers’ professional work and hope that the corrections will make our manuscript suitable for publication in International Journal of Molecular Sciences. We are looking forward to receiving your feedback.

Once again, thank you very much for your constructive comments and suggestions.

Reviewer 2 Report

This manuscript (MS) by Han et al. is a comprehensive review of plant CCCH zinc finger proteins. In this MS, authors discuss the domain structures, subcellular localization, and functions in transcriptional and post-transcriptional regulation via interaction with DNA, RNA, and other proteins. The authors further discuss functions in plant development and response to abiotic stress of CCCH zinc finger proteins. They especially focus on the roles of the proteins in plant abiotic stress tolerance. This MS is well written and covers interesting and important recent findings in plant CCCH zinc-finger proteins. This MS will contribute to enhance insights of plant community about functional roles of plant CCCH zinc-finger proteins. Minor comments and corrections for this MS are as follows.

  1. Line 44 WRKY [9, 29-31] fonts: bold -> plain
  2. Line 46: Ap2/ERF -> AP2/ERF
  3. Line 150: (Wang et al. 2008) -> change citation format according to IJMS guideline
  4. Italicization of gene names: line 317, lines 331-334, line 438, line 489, and etc.
  5. Line 443: in vitro -> italicization

Author Response

Response to Reviewer 2 Comments

Dear Editor and Reviewers,

Thank you very much for your suggestions and critical comments on our paper submitted to International Journal of Molecular Sciences with a revised title of “The Roles of CCCH Zinc- Finger Proteins in Plant Abiotic Stress Tolerance”. We have revised the manuscript according to the comments and suggestions. The main revised parts are marked in red in the revised version. The following is a detailed explanation of how we have complied with the reviewers’ suggestions.

Point 1: Line 44 WRKY [9, 29-31] fonts: bold -> plain.

Response 1: Done. The details are provided in line 50 of the revised version.

Point 2:

Line 46: Ap2/ERF -> AP2/ERF

Response 2: Done. The details are provided in line 52 of the revised version.

Point 3: Line 150: (Wang et al. 2008) -> change citation format according to IJMS guideline.

Response 3: Done. The details are provided in line 156 of the revised version.

Point 4: Italicization of gene names: line 317, lines 331-334, line 438, line 489, and etc.

Response 4: Done. The details are provided in line 365, 378-382, and 465 of the revised version. For line 489, the content was related to biotic stress, so we deleted it in the revised version.

Point 5: Line 443: in vitro -> italicization.

Response 4: Done. The details are provided in line 469 of the revised version.

For the Revised Main Manuscript, please see the attachment.

We have tried our best to improve the manuscript and have made substantial changes and deletions according to the reviewers’ comments. We earnestly appreciate the reviewers’ professional work and hope that the corrections will make our manuscript suitable for publication in International Journal of Molecular Sciences. We are looking forward to receiving your feedback.

Once again, thank you very much for your constructive comments and suggestions.

Round 2

Reviewer 1 Report

The manuscript has been greatly improved. All my queries have been addressed. I believe it can be accepted.